# Single molecule secondary structure determination of proteins through infrared absorption nanospectroscopy

Francesco Simone Ruggeri [1✉], Benedetta Mannini [1], Roman Schmid[1], Michele Vendruscolo [1] & Tuomas P. J. Knowles [1,2✉]

The chemical and structural properties of biomolecules determine their interactions, and thus their functions, in a wide variety of biochemical processes. Innovative imaging methods have been developed to characterise biomolecular structures down to the angstrom level. However, acquiring vibrational absorption spectra at the single molecule level, a benchmark for bulk sample characterization, has remained elusive. Here, we introduce off-resonance, low power and short pulse infrared nanospectroscopy (ORS-nanoIR) to allow the acquisition of infrared absorption spectra and chemical maps at the single molecule level, at high throughput on a second timescale and with a high signal-to-noise ratio (~10–20). This high sensitivity enables the accurate determination of the secondary structure of single protein molecules with over a million-fold lower mass than conventional bulk vibrational spectroscopy. These results pave the way to probe directly the chemical and structural properties of individual biomolecules, as well as their interactions, in a broad range of chemical and biological systems.

[1] Department of Chemistry, University of Cambridge, Cambridge CB2 1EW, UK. [2] Cavendish Laboratory, University of Cambridge, Cambridge CB3 0HE, UK. ✉email: fsr26@cam.ac.uk; tpjk2@cam.ac.uk

The understanding of the structure, function and interactions of biomolecules has greatly advanced through spectroscopic methods[1–4]. In this context a particularly widely applicable approach is vibrational spectroscopy, which represents a sensitive analytical and label-free tool to determine the chemical composition and structural properties of biomolecules[5–7]. For heterogeneous biological systems, however, bulk spectroscopic methods can be challenging to apply since they retrieve information averaged over the ensemble of different molecular species. Thus, there is a compelling need to be able to probe the chemical and structural properties of biological matter at the scale of single molecules, requiring extreme sensitivity in combination with high accuracy and throughput.

Current state-of-the-art nanoscale spectroscopy approaches offer remarkable sensitivity, but there is a tradeoff between sensitivity and ability to relate directly the acquired chemical information into quantitative characterisation of structural properties. Scattering based methods such as tip-enhanced Raman spectroscopy (TERS)[8] and scanning near field optical microscopy (s-SNOM)[9] have enabled the acquisition of chemical information on the nanoscale[10,11] and with single molecule and in some cases even single chemical bond scale[12,13]. However, interpretation of such spectra in terms of direct structural constraints of the biomolecules under study remains challenging and scattering spectra rich in information are subjected to geometric and plasmonic effects, as well as selection rules causing suppression of bulk Raman bands in TERS, such as amide I[14], and thickness-dependent chemical shifts in s-SNOM, which are factors that render quantitative structural analysis less direct than for bulk spectra[15,16]. On the other hand, infrared nanospectroscopy based on thermochemical detection (atomic force microscopy-infrared spectroscopy (AFM-IR)) measures directly the light absorbed by a sample by photothermal induced resonance[17,18]. Thus, the infrared absorption spectra produced are not affected by scattering effects or specific nanoscale selection rules, and as such they are in agreement with conventional bulk results[19–26]. To date, however, the sensitivity of AFM-IR has been limited to the measurements of large (>0.3 μm) and flat (~2–10 nm) self-assembled monolayers or biomolecular aggregates composed of several hundreds of molecules and has not demonstrated the ability to characterise single biomolecules[23,27]. This technological gap is related to the complexity of the AFM-IR thermomechanical response, especially while exploiting the rod-like antenna effect to enhance the electromagnetic field[22,23,27].

Here, to overcome this gap and bring together single protein molecule sensitivity with quantitative structural characterisation from absorption spectroscopy, we unravel the physical principles underlying thermomechanical detection and field enhancement at the nanogap between the metal tip and substrate to achieve up to an order of magnitude increase of the sensitivity of nanoscale absorption spectroscopy. We demonstrate that ORS-nanoIR enables the acquisition of IR absorption spectra and maps from single protein molecules on a time scale of 1 s, with high throughput and signal-to noise ratio (~10–20). In turn, the achievement of this high sensitivity enables the accurate determination of the secondary structure elements of single proteins in the amide band I region, such as α-helices and β-sheets, with similar accuracy than conventional bulk vibrational spectroscopy on samples with over a million-fold larger mass.

## Results

**Off resonance, short pulse and low power nanoIR.** AFM-IR exploits the combination of the high spatial resolution of AFM (~1–10 nm) with the chemical analysis power of IR spectroscopy (Supplementary Note 1)[28]. A scheme illustrating the principle of function of the setup used in our work to acquire infrared absorption maps and spectra from a single monomeric protein (green sphere) is shown in Fig. 1. A tunable quantum cascade (QCL) IR laser is focused on the tip and the protein (Supplementary Fig. 1). If the wavenumber of the exciting laser radiation pulse matches one of the molecular vibrational energy transition levels of the protein, the IR light is absorbed. This absorption causes a thermal heating and expansion of the protein, which is detected by the AFM cantilever[22,27,28]. The IR absorbance at each wavenumber is proportional to the peak-to-peak amplitude of the raw deflection of the cantilever oscillation (Fig. 1a) and to the peak amplitude of its Fourier transform (IR amplitude). To increase the resolution and sensitivity of AFM-IR, we used gold probes and a flat (~0.4 nm roughness) template-stripped gold substrate to exploit the rod-like antenna effect and to enhance the field at the apex of the tip, which is in contact with a single protein (Fig. 1a, Supplementary Fig. 2)[22,27]. In previous studies, the sensitivity has been further increased by matching the laser pulse frequency with one of the eigenvalues of frequency of oscillation of the cantilever in contact with the sample (Fig. 1b). This approach enables the measurement of nanometre scale protein aggregates or flat membrane monolayers with few nanometres thickness[22,23,27]. However, the resonance mode and the rod-like antenna effect cause a complex interaction at the nanogap between the sample, the substrate and the tip, hampering until now the detection of the nanoscale-localised IR absorption of small objects such as single protein molecules. In particular, since a protein has a dimension of only few nanometres, when the sample is not a flat and large monolayer and has dimensions smaller than the apex of the AFM tip, the cantilever is excited by both interactions with gold and the sample (Fig. 1b, c).

To reach single protein molecule detection, we aimed at improving the stability, sensitivity and accuracy of AFM-IR thermomechanical detection by shedding light on the physical principles governing the interaction of the gold-coated silicon probe in contact with the protein and close to the gold substrate. In Fig. 1c, we show the response of the deflection of the cantilever when the tip is placed on the bare gold substrate and on the single protein on the substrate. If the protein has a diameter smaller than the probe diameter (~20-50 nm), a large portion of the gold probe is exposed to the gold substrate at a distance of few nanometres. Thus, the excitation of the gold cantilever over the gold substrate strongly contributes to and competes with the AFM-IR signal arising from the thermomechanical expansion of the protein.

We first aimed at unravelling the strength of interaction between the gold probe and the bare gold substrate (Supplementary Note 1). One may expect that increasing the power of the laser and of the enhanced field should cause an increase of the peak-to-peak deflection cantilever deflection and IR amplitude, and thus instrument sensitivity. Surprisingly, we found that when the tip is placed closed by the substrate, already at the low laser power of 0.35 mW (~0.1% of commercial QCL IR sources) and pulse of 100 ns, the enhancement of the field generates nonlinearity and instabilities in the thermomechanical detection, without any saturation of the signal (Fig. 1d, Supplementary Movies 1–2). The instabilities are caused by a strong interaction of the IR light with the gold tip and substrate, thus fundamentally affecting the quality and reproducibility of the AFM-IR measurements. Consequently, we added a mesh filter (Industrial Netting, USA) at the exit of the laser to our AFM-IR setup to reduce the power of the infrared illumination of almost an order of magnitude. Then, we varied the laser power and pulse to determine the region where the laser-induced cantilever deflection operates in a linear-response regime (Fig. 1d). The presented data demonstrate that in order to have linearity, to maximise the

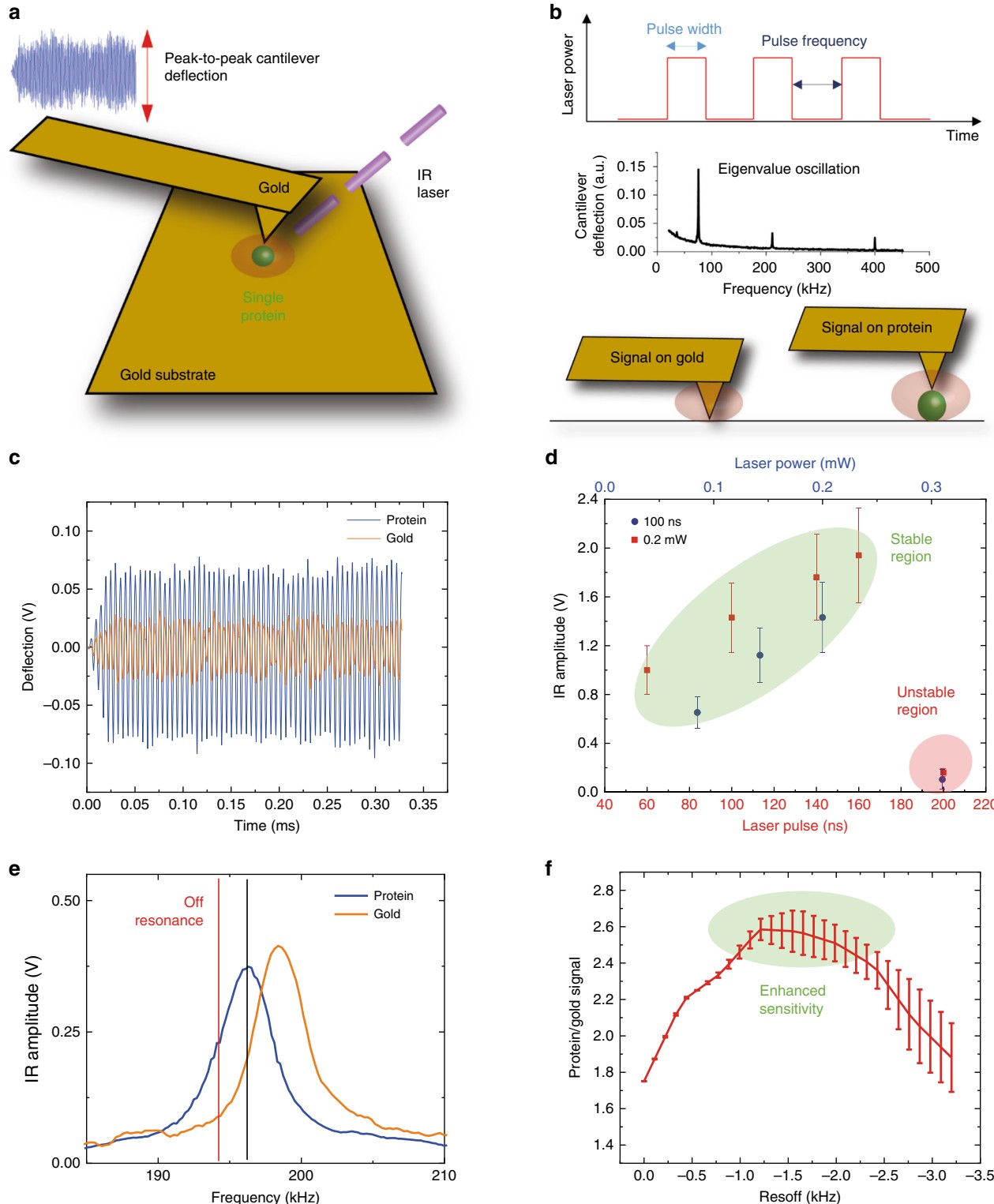

**Fig. 1 Off resonance, short pulse, low power infrared (ORS-nanoIR) absorption spectroscopy of single protein molecules. a** Scheme of the experimental setup, exploiting the rod-like antenna effect at the nanogap between a gold substrate and a gold cantilever. **b** Scheme of the pulsed excitation of the laser (top), of the eigenvalues of oscillation of the cantilever (centre) and of the laser hot spot on gold and on a protein (bottom). **c** Measurement of the cantilever deflection on gold and on a thyroglobulin molecule at a frequency of 194 kHz, power of 0.22 mW and at 1655 cm$^{-1}$. **d** IR amplitude of the cantilever response on the bare gold substrate as a function of laser power and pulse at 1730 cm$^{-1}$; low values define the region of stability of the thermomechanical response (green circle) and high values can cause instabilities and drop of the signal (red circle). **e** IR amplitude as a function of frequency pulse for signal on gold and on a thyroglobulin molecule at a laser power of 0.22 mW and at 1655 cm$^{-1}$. **f** Plot of the ratio between the IR amplitude of the signal on the protein and on the gold as a function of frequency. Since the tip interacts both with gold and protein, the maximum sensitivity in detecting the signal of the protein vs. the signal of gold (green circle) occurs for a laser frequency off resonance (red line) of the maximum of the response on the protein molecule (black line). In all panels, the error bars represent the average + s.d. Source data are provided as a Source Data file.

stability and accuracy of IR absorption detection, it is necessary to excite the sample, probe and substrate system with low power (<0.35 mW) for a pulse width of 100 ns of the laser illumination (green circle in Fig. 1d and Supplementary Note 1). Lower values of pulse width allow to use higher power and vice versa. These conditions of linear response avoid strong excitation of the cantilever, which is of fundamental importance to limit the damage to soft biomolecules during the measurements. These principles can be further generalised for different molecules and vibrations (Supplementary Note 1).

We further aimed at improving the sensitivity of AFM-IR by studying the IR amplitude response on the protein and on the substrate as a function of the pulse frequency of the laser (Fig. 1e, Supplementary Note 1). We start from the observation that the two IR amplitude curves on the protein and on the gold substrate are partially superimposed, for the response of both the second and third eigenvalue of cantilever oscillation (Supplementary Fig. 2). This effect demonstrates that when measuring an object with a diameter smaller than the typical diameter of the AFM tip, the cantilever is also excited by the probe and substrate, an effect which diminishes the sensitivity to the sample itself.

To quantify the influence of the substrate, we studied the ratio between the signal on protein and on gold (protein/gold) as a function of the detuning of the laser pulse frequency (off resonance, $res_{off}$, red dashed line in Fig. 1e) from the frequency corresponding to the maximal IR amplitude (black dashed line in Fig. 1e). We did not find a monotonic trend and the maximal protein/gold signal was found at a $res_{off} \sim -1.5$ kHz (green circle in Fig. 1f) smaller than the contact resonance frequency corresponding to the maximum of the IR amplitude. The protein/substrate signal has a flat response between $-1$ and $-2$ kHz, where it is enhanced by ~70%. These results demonstrate that to minimise the contributions of the probe and substrate for the measurement of the thermomechanical expansion and IR absorption of a single molecule, it is necessary to introduce off-resonance monitoring (detuning on the left of the peak of ~1–2 kHz) of the probe-sample contact resonance frequency and IR absorption measurement.

In summary, we introduced low power, short pulse and off-resonance infrared nanospectroscopy (ORS-nanoIR) for demonstrating the detection of the thermomechanical expansion and IR absorption of samples at the single protein molecule level (Figs. 2–4, Supplementary Note 1). The physical principles and conditions defined by ORS-nanoIR can be similarly extended for all those samples having intrinsic dimensions smaller than the AFM tip diameter, where it is necessary to minimise damage by the tip, as well as the influence of substrate and contaminants.

**Nanoscale chemical imaging of single protein molecules**. We applied the ORS-nanoIR approach to acquire IR absorption spectra and chemical maps from individual protein molecules with similar morphology, but different secondary structures (Supplementary Fig. 4). More specifically, in Figs. 3 and 4 we demonstrate the high-throughput acquisition of chemical information by IR absorption for thyroglobulin (molecular weight 665 kDa, hydrodynamic radius ~8 nm) and apoferritin (molecular weight 443 kDa, hydrodynamic radius ~6 nm) with high signal-to-noise ratio (SNR)[29,30]. First, we simultaneously acquired by ORS-nanoIR correlated maps of three-dimensional (3D) morphology (Fig. 2a), IR absorption (Fig. 2b) and nanomechanical properties (Fig. 2c) of thyroglobulin molecules on a gold-coated substrate. Then, we measured a cross-section of the three signals at the same position on single protein species. The height and the IR absorption of the proteins are excellently correlated (Fig. 2d). In Fig. 2c we see that that the contact resonance frequency decreases

in correspondence of the protein in both the topographic height and IR amplitude, indicating that the system is correctly tracking the contact resonance and thus the IR absorption.

In order to prove single protein chemical sensitivity, we performed a comparative analysis of the volume of the species measured by AFM with the known experimental 3D structure of both apoferritin and thyroglobulin[29,30]. First, we characterised the roughness of the gold substrates before and after the deposition of the proteins to demonstrate the capability to measure single protein topography (Supplementary Fig. 5). While AFM can directly probe the height of the protein with a sensitivity in the order of angstroms[31], the cross-sectional measurement of the lateral dimensions of the proteins are significantly overestimated because of tip convolution effects (Fig. 2e). Thus, we calculated the deconvoluted cross-sectional dimensions of the proteins on the surface to measure their volume (Methods, Supplementary Figs. 6 and 7)[29,30]. Then, we compared the measured deconvoluted volume of the protein species on the surface with the estimated theoretical volume of apoferritin and thyroglobulin (Fig. 1f, Supplementary Fig. 6), which was calculated from the known crystal structures and the measured hydrodynamic radius of the two proteins (Supplementary Fig. 8). The volume of the protein species measured by AFM and calculated from the crystal structures were in excellent agreement with the volume of a single protein and much smaller than the volume of two proteins (Supplementary Fig. 7). Thus, within the volume of the morphological shapes and topography observed, it is possible to have only a single protein. We proved independently this result by measuring the AFM volume of thyroglobulin by a conventional AFM setup equipped with a sharper probe (nominal radius ~8 nm) than the gold probes used for AFM-IR (nominal radius ~30 nm) (Supplementary Fig. 7). We observed on the surface also abundant protein species with a smaller volume than a single thyroglobulin molecule (Fig. 2). A thyroglobulin protein is composed by two identical subunits and indeed the volume of these smaller species is compatible with the one of the two identical monomeric subunits forming a thyroglobulin molecule. The presence of individual subunits and smaller protein species in solution is independently confirmed by dynamic light scattering (DLS) (Supplementary Fig. 8). We can conclude that AFM-IR chemical imaging enable to discriminate monomeric thyroglobulin from its subdomains, thus demonstrating sub-protein sensitivity. We finally used the sharp edge method and first derivative analysis to demonstrate a spatial resolution, which was in the order of ~10 nm (Supplementary Fig. 9).

**Secondary structure determination of single protein molecules**. After the acquisition of the nanoscale-resolved maps, we placed our probe on the top of a single thyroglobulin molecule (Fig. 3a, b, Supplementary Fig. 10) and apoferritin (Supplementary Fig. 11) to demonstrate the acquisition of IR absorption spectra in the characteristic regions of proteins corresponding to the amide bands I, II and III (Fig. 3c). The amide I band is the most frequently used band to infer the secondary structure of peptides since it arises in large part (80%) from the backbone $C=O$ stretching vibration, which is strictly related to the protein secondary and quaternary structure[19]. In Fig. 3a, we show the 3D structure of thyroglobulin.

After protein deposition, the surface roughness increases because of the presence of secondary solutes (buffer, glycerol) as well as possible residual protein material (Supplementary Fig. 5). Furthermore, during measurements, the probe can be as well contaminated. In Fig. 3b, a morphology map of a single protein molecule is shown and the crosses indicate the location of acquisition of the IR absorption spectra on the protein (blue) and

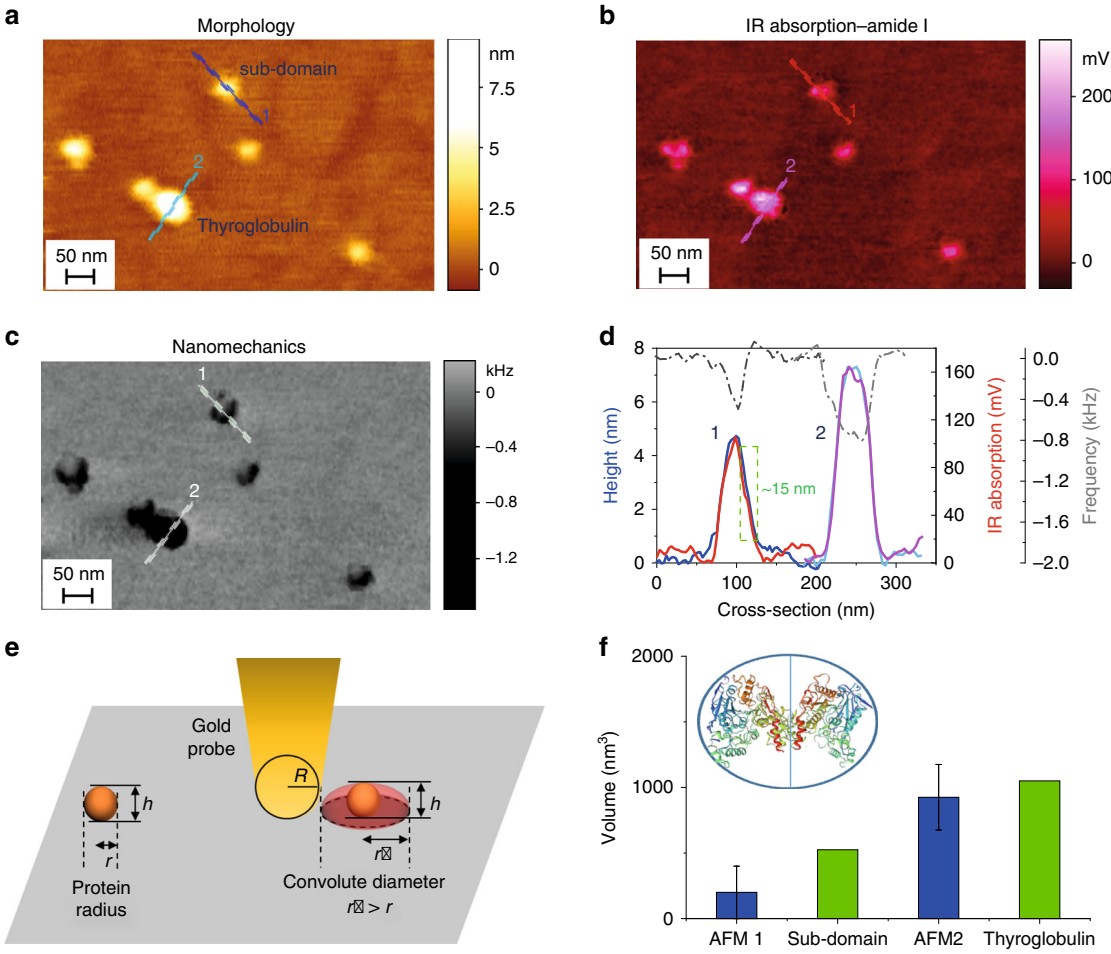

**Fig. 2 Simultaneous morphological, chemical and mechanical imaging by IR absorption of single protein molecules. a** 3D morphology map of thyroglobulin molecules. **b** IR absorption image at 1655 cm$^{-1}$ (amide band I) and **c** contact resonance frequency map at 1655 cm$^{-1}$ (laser power: 0.24 mW; pulse: 100 ns). **d** Cross-sectional correlation of morphology, IR absorption and nanomechanical properties for the protein species in the map. **e** An overestimate of the lateral radius $r$ and of the volume of the single proteins in the AFM-IR map is caused by the finite geometrical shape and radius of the tip[31,32]. **f** Inset of the graph is the structure of a thyroglobulin molecule, which is composed by two identical subdomains[29]. We demonstrate single protein sensitivity by comparing the known volumes of a thyroglobulin molecule (blue) and its subdomain (green)[30], with the deconvoluted volume of each single protein species measured on the surface by AFM (blue, the error on the deconvoluted volume is calculated by varying the radius of the tip between $R$ ~30 and 40 nm) (Supplementary Fig. 6)[31]. The volume of a single protein measured by AFM (blue) is in excellent agreement with the values of the volume of a single subdomain and thyroglobulin. Source data are provided as a Source Data file.

on the gold substrate (grey) in the spectroscopic range of protein amide bands I, II and III (Fig. 3c, d). To remove the contribution of the contaminants on the surface and on the probe, we acquired the spectra of a protein and on its surrounding substrate and then we subtracted the spectrum acquired on the protein from the spectrum on its surrounding (Fig. 3d). Thus, even if on the tip or substrate are present residual contaminants, such as eventual protein fragments, they have been successfully subtracted from the spectra. In order to decrease the level of noise, we average three subtracted spectra of the same protein (Fig. 3e). The average spectrum is further baselined and smoothed by a Savitzky–Golay filter and it shows clearly the characteristic amide bands I, II and III of protein above the noise (Fig. 3f).

Previous results reported in the literature have shown that apoferritin has an α-helical content of 75% (Fig. 4a)[33], while thyroglobulin has contributions from α-helix and coils (55%), β-turn (17%) and intramolecular β-sheet structures (28%) (Fig. 4b)[29]. Then, we considered the average spectra on three different spectra of single molecules of apoferritin and thyroglobulin measured by ORS-NanoIR (Fig. 4c, d, Supplementary Figs. 10 and 11). We calculated the second derivatives of the

spectra to de-convolve and integrate the major secondary structural contribution to the amide band I (Fig. 4d, e, Supplementary Fig. 12)[19,34]. As demonstrated in Fig. 4f, g, the secondary structure determined from the AFM-IR spectra at the single protein molecule level, for both apoferritin and thyroglobulin, was in excellent agreement with previous results and with the quantification of the secondary structure from our bulk FTIR spectra (Fig. 4d, e, Supplementary Note 2).

## Discussion

We have demonstrated that the ORS-nanoIR method enables the acquisition of IR absorption spectra from single protein molecules, with a molecular weight of ~443 kDa and a stiffness of ~30 MPa, to determine their secondary structures. A protein has a typical linear expansion coefficient $\alpha$~0.001 K$^{-1}$ (ref. [35]). For a temperature change of $\Delta T$~5–7 K[22,27], we have a linear expansion of ~0.5–0.7%. For a protein with a diameter of 12-16 nm, we have a linear expansion of ~0.06-0.12 nm. For a thermal expansion of ~0.1 nm, a Young's modulus of a protein of ~30 MPa, a tip radius of ~25 nm and an indentation of ~1 nm (<10% of protein height), we estimate a measured force $F$~0.3 nN, which is comparable

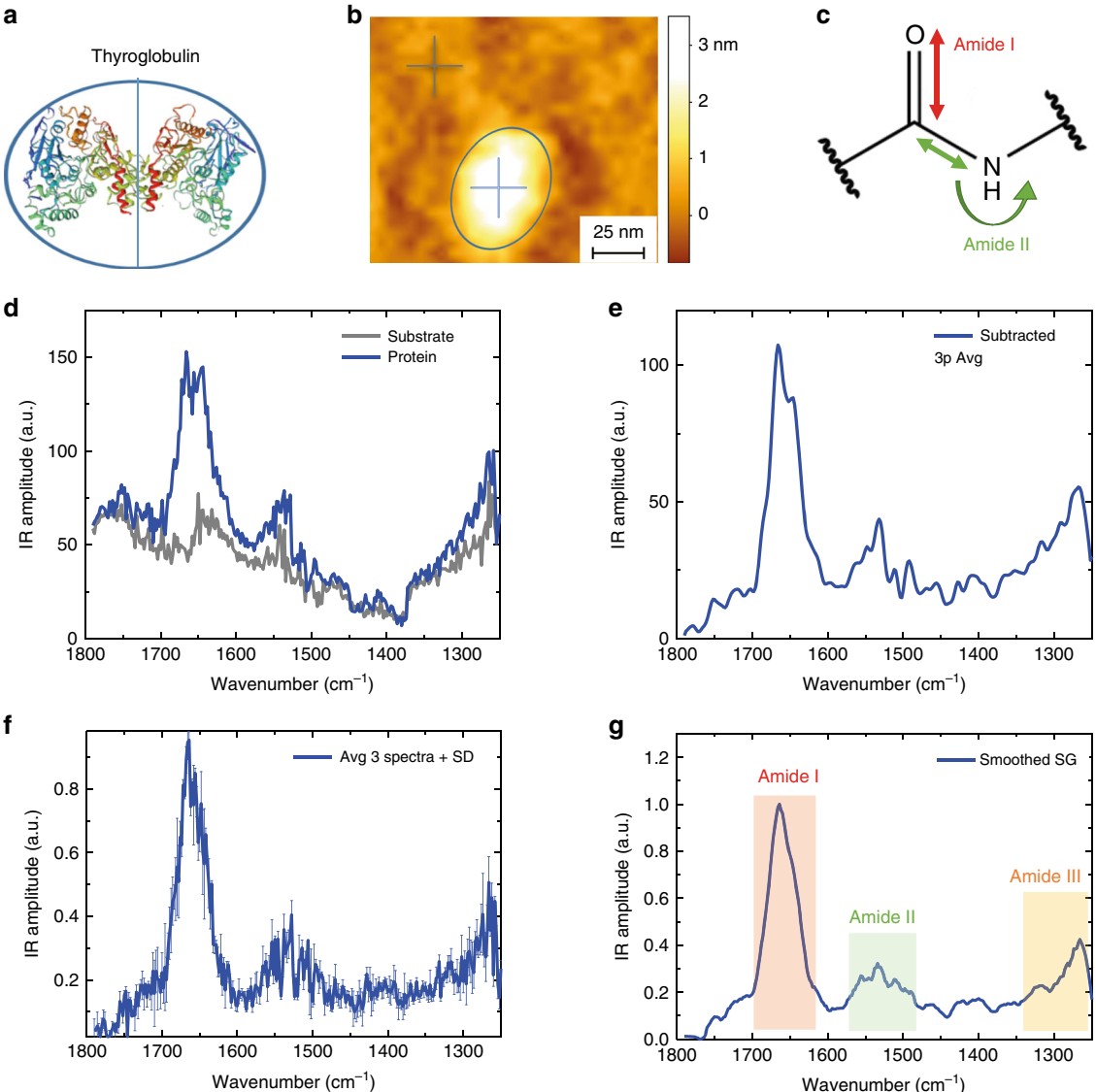

**Fig. 3 ORS-nanoIR absorption spectra of single protein molecules. a** Thyroglobulin structure[29]. **b** Morphology map of a single protein (blue circle).
**c** Schematic of the chemical bond vibration leading to the amide band I (80% C = O stretching, 10% C–N stretching; 10% N–H bending), amide band II
(60% N–H bending and 40% C–N stretching)[19]. **d** IR absorption spectra acquired on the protein (blue cross), on gold (grey cross) (laser power: 0.3 mW;
pulse: 60 ns). **e** Protein–substrate subtraction spectrum with adjacent averaging of three points. **f** Average of three different subtracted spectra on the
same protein with standard error. **g** Savitzky–Golay smoothed (second order, 13 pt) and baselined spectrum in amide bands I, II, III. The error bars represent
the average + s.d. Source data are provided as a Source Data file.

with previous results in literature[27]. In order to improve the
sensitivity of the conventional implementation of AFM-IR to
reach single protein sensitivity, we needed to reduce the power
and pulse excitation of the gold cantilever over the gold substrate
to drive more efficiently the cantilever response, also empowering
the possibility to study very soft samples, such as single protein
molecules, preserving their conformations, and increase the signal
of protein over the signal arising from the substrate and the tip of
~70% by measuring off resonance of excitation.

In summary, the ORS-nanoIR method enables the direct
acquisition of absorption infrared spectra and maps of single
protein molecules and opens a new window of observation on the
chemical and structural properties of individual biomolecules.
Future improvements in the technology, such as of coating
metallic material and sharpness of the probes, can pave the way
towards single oscillator detection. Furthermore, developments of
the sensitivity of this technique in physiological environments

will offer fruitful avenues for the study of biomolecules and their
interactions in native and liquid environments of physiological
relevance for a wide range of biomedical and biotechnological
applications.

## Methods

**Preparation of monomeric proteins**. We have chosen apoferritin and thyr-
oglobulin since they have been extensively characterised in literature and they are
used as calibration tools to study the biophysical behaviour of unknown mono-
meric proteins[36]. We used identical conditions of the buffer solutions as in pre-
vious protocols in literature[30], which enable to preserve the monomeric state of the
two proteins. Apoferritin was purchased by Sigma Aldrich (Catalogue number
A3660) supplied as a solution at 25 mg ml$^{-1}$ in 50% glycerol with 0.075 M NaCl.
Thyroglobulin was purchased by Sigma Aldrich (Catalogue number T9145) as
powder and dissolved in buffer (50 mM Tris-HCl, pH 7.5, with 100 mM KCl).
Before all measurements, the proteins were centrifuged at 21,130 r.c.t. at 4 °C for 5
min and then filtered by using a 22 μm filter. We further demonstrated the
monomeric state of our proteins in solution by DLS (Supplementary Fig. 8).

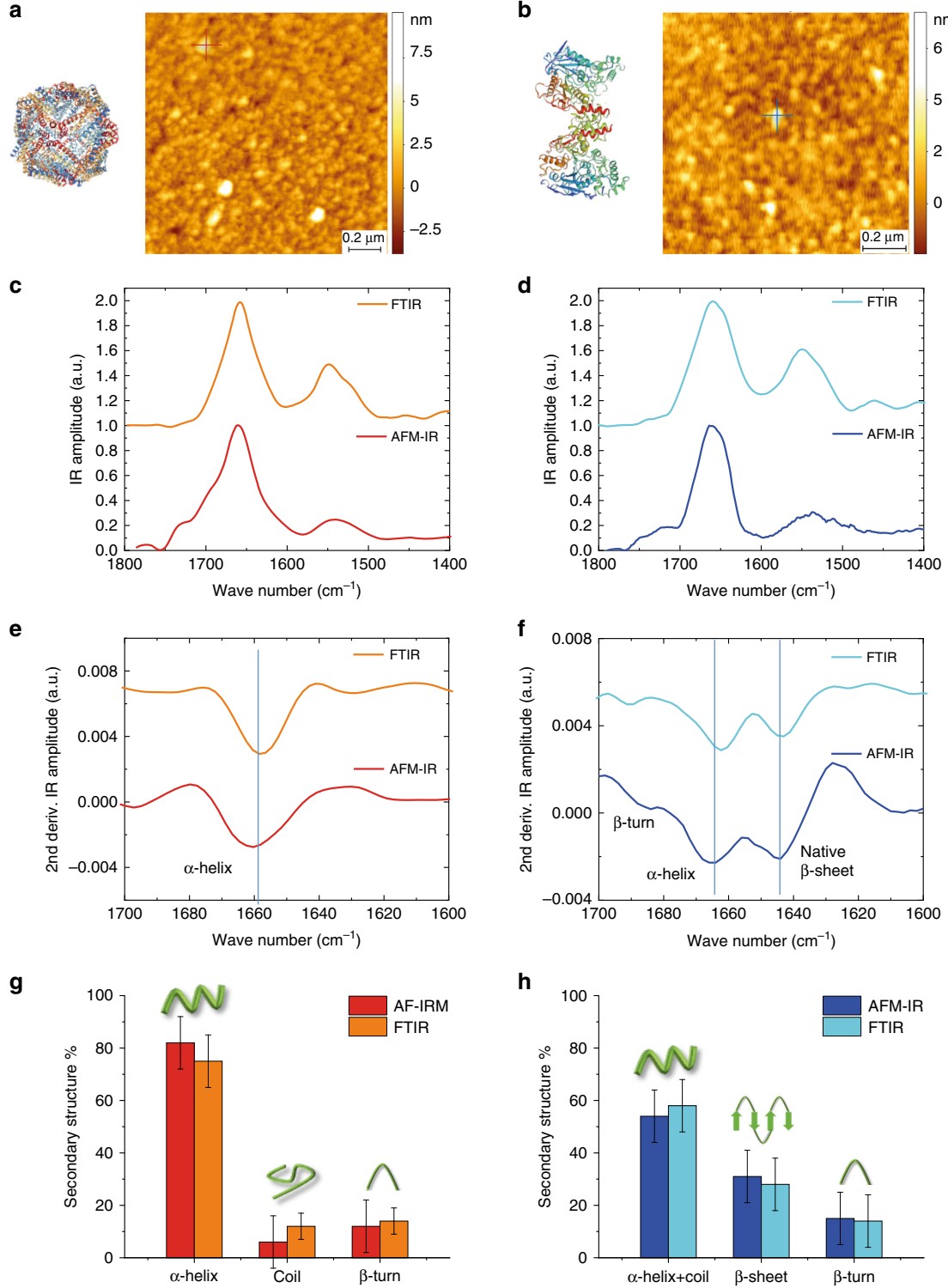

**Fig. 4 Secondary structure determination from ORS-nanoIR absorption spectra.** Structures and morphologies of **a** apoferritin[33] (red cross) and **b** thyroglobulin (blue cross)[29]. **c**, **d** Average IR spectrum of three different protein molecules and comparison with bulk IR spectra. **e**, **f** Deconvolution of the structural contributions in the amide band I by second-derivative analysis (Savitzky–Golay smoothed, second order, 7 pt) and comparison with bulk FTIR spectra. Comparison of the measured secondary structure by bulk FTIR and single-molecule AFM-IR of **g** apoferritin and **h** thyroglobulin. The experimental error arises from the sum of the s.e. of the mean of three spectra averaged and the fit convolution fit error. Source data are provided as a Source Data file.

**FTIR measurements**. Attenuated total reflection infrared spectroscopy (ATR-FTIR) was performed using a Bruker Vertex 70 spectrometer equipped with a diamond ATR element. Spectra were acquired with a concentration of 20 µM apoferritin and 12 µM thyroglobulin. The resolution was $4\,cm^{-1}$ and all spectra were processed using OriginPro software. The spectra were averaged (3 spectra with 412 co-averages), smoothed applying a Savitzky–Golay filter (second order,

9 points) and then the second derivative was calculated applying a Savitzky–Golay filter (second order, 11 points).

**Circular dichroism measurements**. CD experiments were carried out using a Jasco J-810 spectropolarimeter equipped with a Peltier holder. CD spectra were measured at a protein concentration of 1 and 0.25 µM of apoferritin and

thyroglobulin. Measurements were performed with a scanning speed of 50 nm min$^{-1}$ and a data pitch of 0.5 nm at 20 °C. Spectra were averaged from 20 scans and smoothed using the "means-movement" smoothing procedure implemented in the Spectra Manager package. The contribution of buffer was subtracted from experimental spectra. Mean ellipticity values per residue (MRE) were calculated as $= \frac{\theta_{obs}}{10 n c l}$, where $l$ is the path length (0.1 cm) and $n$ is the number of residues and $c$ the protein concentration expressed in mol cm$^{-3}$.

**Dynamic light scattering**. DLS measurements were performed at 25 °C using the Malvern Zetasizer Nano S instrument (Malvern, Worcestershire, England) equipped with a Peltier temperature controller. Measurements were acquired at the concentration of 0.05 mg mL$^{-1}$ in the buffer conditions described above.

**AFM-IR measurements, maps treatment and analysis**. Analysis by nanoIR2 (Anasys Instrument, USA) was performed on atomically flat gold substrates with a nominal roughness of 0.36 nm (Platypus Technologies, USA)[37]. The substrate roughness and chemical response were also characterised by AFM-IR in the Supplementary Fig. 2. The root mean square roughness of the AFM maps was measured by SPIP (Image metrology, Denmark).

To prepare the protein samples, an aliquot of 10 µl at 4 µM concentration was deposited on the flat gold surface for 10 s to reduce mass transport phenomena during drying. Successively, the droplet was rinsed by 1 ml of Milli-Q water and dried by a gentle stream of nitrogen. The morphology of the protein samples was scanned by the nanoIR microscopy system, with a rate line within 0.1–0.5 Hz and in contact mode. All AFM maps were acquired with a resolution between 1 and 5 pixels nm$^{-1}$. A silicon gold-coated PR-EX-nIR2 (Anasys, USA) cantilever with a nominal radius of ~30 nm and an elastic constant of about 0.2 N m$^{-1}$ was used. To use gold–gold rod-like antenna effect, the IR light was polarised perpendicular to the surface of deposition. The AFM images were treated and analysed using SPIP software. The height images were first-order flattened, while IR and stiffness-related maps were only flattened by a zero-order algorithm (offset). Spectra were collected with a laser wavelength sampling of 2 cm$^{-1}$ with a spectral resolution of 0.1 cm$^{-1}$ and 256 co-averages, within the range 1250–1800 cm$^{-1}$. The spectra were acquired at a speed of 100 cm$^{-1}$ s$^{-1}$. Furthermore, to cover our spectral range two QCL lasers were employed, which added a time to switch between the two lasers chip of approximately 300–500 ms. Thus, the acquisition time for our spectral range is shorter than 1 s. All spectra and maps were acquired at the same power of background laser power, between 0.13 and 0.35 mW and with a pulse width between 40 and 100 ns. Since the spectral background line shape slightly depends on laser power, the spectra were normalised by the QCL emission profile at the same power (Supplementary Fig. 1). Spectra were analysed using the microscope's built-in Analysis Studio (Anasys) and OriginPRO; structural contributions were calculated by second derivative analysis and deconvolution of the amide band I (Supplementary Fig. 12)[19,20,34]. The second derivatives were smoothed by a Savitzky–Golay filter (second order, 7 pt).

All measurements were performed at room temperature under controlled nitrogen atmosphere with residual real humidity below 5%. Both spectra and images were acquired by using phase-locked loop (PLL) tracking of contact resonance, the phase was set to zero to the desired off-resonant frequency on the left of the IR amplitude maximum, and tracked with an integral gain $I = 0.1$ and proportional gain $P = 5$ (refs. [38–40]).

**Atomic force microscopy**. Atomic force microscopy was performed on bare mica substrates. To prepare the protein samples, an aliquot of 10 µl at 4 µM concentration was deposited on the flat gold surface for 10 s. AFM maps were acquired by means of a Multimode VIII (Bruker, USA) and a NX10 (Park systems, South Korea) operating in tapping mode and equipped with a silicon tip (µmasch, 2 N m$^{-1}$) with a nominal radius of ~8 nm. Image flattening and single aggregate cross-sectional dimension analysis were performed by SPIP (Image Metrology) software.

**Determination of protein volume**. In order to prove that the protein species on the surface in Figs. 2–4 are single apoferritin and thyroglobulin molecules, we performed a comparative analysis of the volume of the species measured by AFM with the known experimental 3D structure of the two proteins (Fig. 2, Supplementary Fig. 6). Thus, we have first calculated the deconvoluted cross-sectional dimensions and the volume of the individual protein species, where AFM-IR spectra where acquired. Then, we have compared the deconvoluted AFM volume with the volume calculated from crystal structure and hydrodynamic radii measured by DLS (Supplementary Fig. 8)[29,30].

To calculate the deconvoluted volume of the proteins, we first measured their 3D cross-sectional dimensions from the AFM maps. While the measurement of cross-sectional height by AFM has a resolution of a fraction of a nanometre and is not strongly affected by the tip geometry, the shape of the tip is the primary determinant of the cross-sectional dimensions and of the lateral resolution[31]. Since the apical radius of the gold probe used by AFM-IR is larger (~30 nm) than the protein radius (~6-8 nm), the shape of the protein is affected by significant lateral broadening, which is also known as a convolution effect. We quantified the deconvoluted radius $r'$ of the aggregates as described previously in the literature[31,32], using in first approximation the formula $r = \frac{d^2}{16 R_T}$, where $d$ is the

convoluted diameter of the protein measured by AFM and $R_T$ is the nominal radius of the probe. Already before any measurement, the nominal value of the radius of tip (~30 nm) is the best possible value for its sharpness and manufacturing variations are present. Since the measurements are performed in contact mode, and tip degradation and contamination occurs, we considered the variation of the radius of the gold AFM-IR probe varying between 35 ± 5 nm, to calculate the experimental error on the volume calculation. While, in the case of the less-invasive non-contact mode measurements by AFM in Supplementary Fig. 7, we considered the radius of the probe varying between 10 ± 2 nm (nominal value 8 nm).

Then, we estimated the volume of apoferritin and thyroglobulin from the known crystal structures and the measured hydrodynamic radius of the two proteins (Supplementary Fig. 8). We considered apoferritin as a sphere with a radius of ~6.1 nm, while we considered thyroglobulin an ellipsoid with longitudinal axis with a radius of ~10 nm and an average equatorial axis of radius of ~5 nm (corresponding to a hydrodynamic radius of ~8 nm). We multiplied the obtained value by two to consider the maximal theoretical volume of a dimer of apoferritin and thyroglobulin.

## Data availability

All data needed to evaluate the conclusions in the paper are present in the paper and the Supplementary Information file. The source data underlying Figs. 1–4 and Supplementary Figs 1, 2, 4–8, 10, 11 are provided as a Source Data file. Other data are available from the corresponding authors upon reasonable request.

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

## Acknowledgements

We thank Darwin College and Swiss National Fondation for Science (SNF) for the financial support (grant number P2ELP2_162116 and P300P2_171219). The research leading to these results has received funding from the European Research Council under the European Union's Seventh Framework Programme (FP7/2007-2013) through the ERC grant PhysProt (agreement no. 337969) and from the Wellcome Trust under the Collaborative Awards in Science scheme.

## Author contributions

F.S.R. and T.P.J.K. conceived the project. F.S.R., R.S. and B.M. performed the experiments. F.S.R. analysed the data. F.S.R., M.V. and T.P.J.K. wrote and commented the article.

## Competing interests

The authors declare no competing interests.
