## [Peer Review File · Nature Communications]

Reviewers' comments:

Reviewer #1 (Remarks to the Author):

The manuscript by Ruggeri and co-authors reports on structural determination of single proteins using AFM-IR. Specifically, the authors showed that using this highly sensitive technique they could probe secondary structure of thyroglobulin and apoferritin determining amounts of α -helix, β -turns and coil in their structures. The manuscript is well-written and should eventually be accepted to the publication. However, prior to it, the authors must address the most important question of their work that is not entirely clear. How do the researchers know that they dealt with one protein molecule rather than with a protein aggregate. The reviewer suggests to use alternative analytical technique, such as DLS, to convince the audience that single molecules of the mentioned above proteins were studied. Once demonstrated, the reviewer suggests to accept the manuscript.

Reviewer #2 (Remarks to the Author):

The article by Ruggeri et al addresses the investigation of isolated large biomolecules by the photothermal AFM-IR technique. For increased sensitivity a slightly off-resonant measurement modus was used in a study of isolated apoferritin and thyroglobulin molecules on a gold substrate. Conclusions on the secondary structure of the investigated proteins are drawn with the interpretation of measured AFM-IR and supplementary CD spectra.

Although the thyroglobulin molecules are with dimensions of about 8 nm x 10 nm x 5 nm quite large, to my knowledge this is the first single molecule study by an mid-infrared spectroscopic technique. As such new experimental IR spectroscopic possibilities give access to the analysis of structural and chemical properties of single large biomolecules and macromolecules, they could be in particular relevant for biomedical research. I recommend the publication of the article after reasonable revision.

Major points:

I) How do you know that really only one molecule was measured? Please comment on the tip convolution effects in the text.

II) It is claimed that physical properties of thermomechanical detection and field enhancement are unraveled, where?

III) Potential future possibilities of the technique should be discussed better. In this context: e.g., could the internal structure of the large biomolecules be studied? Why is it not studied yet? It is written that AFM-IR has spatial resolutions from about 1 to 10 nm. This should be maybe sufficient to investigate e.g. the two parts of the homodimer thyroglobulin?

IV) What was the spatial resolution in the presented experiments? Please add. If the spatial resolution was below 10 nm, why can the molecules not be studied without probing any parts of the bare gold substrate? Please comment.

V) With respect to the used measurement modus, why didn't the authors use the shown higher order resonance at around 400 kHz in Fig. 1b? For this one protein and gold resonance should be better separated than for the used resonance frequency at 194 kHz.

VI) Make a statement if the spectrum of the substrate is influenced by the QCI emission profile related background as shown in Fig S1. How does this background and low sensitivity in particular influence the amide 2 band shape in the spectra of the proteins, can this be related to the significant differences between FTIR and AFM-IR spectra as seen in Figure 4. If not, what is the reason?

Minor:

- Introduction: Thus, the IR absorption ... are not affected by plasmonic effects.

I guess plasmonic resonances are meant here?. Please clarify.

- Result sections:

Add the technical details of the used mesh filter.

- Caption Figure 1: ... high values can cause instabilities ..
- Comment: .. Maybe the signal is saturated?..
- Figure 2 b: as it can be barely identified or mistaken please mark the measurement line by an other colour, e.g. green.
- Materials section:
What means a 20 μM of apoferritin.. ; solution?
- Please control referencing of Ref. [32] after (Figure 4b) in the Results section.
- A reference for the assignments used in the deconvolution of the amide I band (supplementary part) is missing.
Supplementary:
- 1st sentence: typo: ... we aimed at we aimed
- Line 33: 10 mW is given as the lower limit for typical power of the laser. This sounds quite high for the study of an organic samples.
- Line 56: contact resonance at higher angle?
- A reference for the assignments used in the deconvolution of the amide I band (supplementary part) is missing.

Reviewer #3 (Remarks to the Author):

Manuscript untitled: « Single Molecule Secondary Structure Determination of Protein through Off-Resonance and Short Pulse Infrared Absorption Nanospectroscopy »

I don't recommend its publication in nature communication: it's not innovative enough and the format is not appropriated to this technical article.
Here are some reasons and the list is not comprehensive.

Here are my comments regarding the manuscript:

General comments:

The authors introduce off-resonance, low power and short pulse AFM-IR to acquire infrared absorption spectra of thyroglobulin at the single molecule level, with a high signal-to-noise ratio. Without any doubt, it's essential to the community to have access to new protocols to improve the sensitivity and reliability of this technique but it's not innovative enough to be published in a journal like nature communication.

I don't want to minimize the interest of this publication. Since the analysis of single protein using AFM-IR is a real challenge, I suggest to the authors to publish it in a more specialized journal where they can bring a real discussion, develop the different ambiguous points and illustrate it with more figures. This way, I think it should be appreciated to its full value by giving more details (by merging the main text and the supplementary documents) and complementary information and by highlighting the results with more proteins.

-You said "To date, however, the sensitivity of AFM-IR has been limited to the measurements of large ($>0.3 \mu\text{m}$) and flat ($\sim 2-10 \text{ nm}$) self-assembled monolayers or biomolecular aggregates composed of hundreds of molecules and has not demonstrated the ability to characterise single molecules": check other publications and update your bibliography.

Some points are a little bit confusing:

- Figure 1 d) is really difficult to understand. Are you sure of your legend (blue dots 100ns ? red dots 7%) I don't understand the link.
- Figure e): if your system is well optimized (in term of power and pulse width) the amplitude of the resonance of the substrate should be lower than the one of the protein. It's not the case. Why?
- in the caption of the figure 1 "(e) IR amplitude as a function of frequency pulse for signal on gold and on a thyroglobulin molecule at a laser power of 7% and at 1655 cm^{-1} " the power used for the

experiments is 7 % but in the text it's written p5 that "already at the low laser power of 0.5 mW (1-5% of commercial IR sources), the enhancement of the field causes non-linearity and instabilities" may be I have missed something but the power used for the experiments should be lower than 7% ? I suppose it's 7% of the power already reduce after the addition of the filter. I think it's better to use only mW and not %.

-Then figure 2 and 3 no power nor pulse width is indicated.

-What about the discrepancies observed between FTIR results and AFM-IR? For example why is your amide II lower? May be there is a physical reason behind it: discuss it.

But my main concern is about the interpretation of the local IR spectra: "Apoferritin has an α -helical content of 75% (Figure 4a),³⁴ while thyroglobulin has major contributions from α -helix and coils (55%), β -turn (17%) and intramolecular β -sheet structures (28%) (Figure 4b)." How you can propose such repartition without showing a curve fitting? the reference extract from FTIR curve fitting (reference) should be in the main document.

Reviewer: 1

Reviewer: The manuscript by Ruggeri and co-authors reports on structural determination of single proteins using AFM-IR. Specifically, the authors showed that using this highly sensitive technique they could probe secondary structure of thyroglobulin and apoferritin determining amounts of α -helix, β -turns and coil in their structures. The manuscript is well-written and should eventually be accepted to the publication.

Answer: We would like to thank the reviewer for these very positive comments on the quality and significance of our study.

Reviewer: However, prior to it, the authors must address the most important question of their work that is not entirely clear. How do the researchers know that they dealt with one protein molecule rather than with a protein aggregate. The reviewer suggests to use alternative analytical technique, such as DLS, to convince the audience that single molecules of the mentioned above proteins were studied. Once demonstrated, the reviewer suggests to accept the manuscript.

Answer: We thank the reviewer for making this excellent point, which has helped to raise the quality of the manuscript significantly. We now demonstrate the monomeric state of our proteins in solution by DLS (**Supplementary Fig. S8**). Furthermore, the volume of the proteins calculated from the measurement of the hydrodynamic radii was in excellent agreement with the deconvoluted volume measured by AFM-IR

Reviewer: 2

Reviewer: The article by Ruggeri et al addresses the investigation of isolated large biomolecules by the photothermal AFM-IR technique. For increased sensitivity a slightly off-resonant measurement modus was used in a study of isolated apoferritin and thyroglobulin molecules on a gold substrate. Conclusions on the secondary structure of the investigated proteins are drawn with the interpretation of measured AFM-IR and supplementary CD spectra.

Although the thyroglobulin molecules are with dimensions of about 8 nm x 10 nm x 5 nm quite large, to my knowledge this is the first single molecule study by an mid-infrared spectroscopic technique. As such new experimental IR spectroscopic possibilities give access to the analysis of structural and chemical properties of single large biomolecules and macromolecules, they could be in particular relevant for biomedical research. I recommend the publication of the article after reasonable revision.

Answer: We would like to thank the reviewer for these very positive comments, and for highlighting the importance and broad impact of our results. Following her/his comments, as detailed below, the manuscript has significantly improved.

Major points:

Reviewer: I) How do you know that really only one molecule was measured? Please comment on the tip convolution effects in the text.

Answer: Following the question of the reviewer, we comment now in the manuscript of tip convolution. In addition, we further analysed of our AFM-IR data (**Fig. 2** and **Fig. S5-6**), and we acquired new high-resolution AFM measurements (**Fig. S7**). To demonstrate the measurement of a single protein, we calculated the deconvoluted radii and volume of the protein species measured by AFM-IR and compared them with the volume of apoferritin and thyroglobulin calculated from crystal structures and from our new measurements of hydrodynamic radius by DLS (**new Fig.2** and **SI Figs. S5-8**). For both apoferritin and thyroglobulin, the volume measured by AFM-IR is in excellent agreement with the calculated volume from the crystal structure of a single protein, and it is significantly smaller than the theoretical volume of a dimer. Thus, within the AFM volume where the spectra were acquired, only a single protein can be present.

Reviewer: II) It is claimed that physical properties of thermomechanical detection and field enhancement are unraveled, where?

Answer: Following the question of the reviewer, we have clarified the results and conclusions section of the manuscript. In summary, we unravelled the strength of interaction at the nanogap between the gold probe and substrate and in particular its influence on: i) the stability and linearity of the thermomechanical detection, which we improved by reducing and balancing the power and pulse of laser excitation to drive more efficiently the cantilever response; ii) the thermomechanical detection arising from the tip and substrate, which competes with the signal arising from a single protein; this approach avoided the damage of the protein by large oscillations of the cantilever and enabled to increase the signal of protein over the signal of the substrate of ~70% by measuring off-resonance of excitation.

Reviewer: III) Potential future possibilities of the technique should be discussed better. In this context: e.g., could the internal structure of the large biomolecules be studied? Why is it not studied yet? It is written that AFM-IR has spatial resolutions from about 1 to 10 nm. This should be maybe sufficient to investigate e.g. the two parts of the homodimer thyroglobulin?

Answer: We comment now on future possibilities in the conclusions of the manuscript. While spatial resolution of AFM is between 1-10 nm, the current spatial resolution of AFM-IR is in the order of ~10 nm and as such does not allow to enquire sub-domain structure. However, further improvements in the sharpness and coating of metal probes used offer a fruitful avenue to address this challenge in the future.

Reviewer: IV) What was the spatial resolution in the presented experiments? Please add. If the spatial resolution was below 10 nm, why can the molecules not be studied without probing any parts of the bare gold substrate? Please comment.

Answer: We evaluated rigorously the spatial resolution on the current study in the **new Fig. S9**. In our study we obtained a lateral spatial resolution of ~10-15 nm, arising mainly from pixelation and tip-convolution.

Reviewer: V) With respect to the used measurement modus, why didn't the authors use the shown higher order resonance at around 400 kHz in Fig. 1b? For this one protein and gold resonance should be better separated than for the used resonance frequency at 194 kHz.

Answer: We fully agree with the reviewer that at higher resonance around 400 kHz the protein and gold resonance peaks are better separated; however, the peaks are also broader and the amplitude of resonance of the protein at this peak was lower (**new SI Fig. 2**). For the reasons above, we based our discussion on the second resonance, but it similarly applies to the third resonance.

Reviewer: VI) Make a statement if the spectrum of the substrate is influenced by the QCL emission profile related background as shown in Fig S1. How does this background and low sensitivity in particular influence the amide 2 band shape in the spectra of the proteins, can this be related to the significant differences between FTIR and AFM-IR spectra as seen in Figure 4. If not, what is the reason?

Answer: We state now in the SI text that our spectra are normalised by the QCL emission profile. We fully agree with the reviewer; the smaller value of the Amide II mode is related to the low laser power in this region. The laser power in the Amide II is three times lower than in the Amide I, as well as the Amide II is absorbing IR light approximately twice less than the Amide I, thus the detection of Amide II is at the limit of instrumental sensitivity. While, it has been recently reported that, for ordered biomaterials and protein embedded in membranes, the lower absolute value of this band could be attributed to polarisation effects (*Giliberti, Nano Letters, 2019*), we believe polarisation effects are not likely for isotropic proteins.

Minor points:

Reviewer: Introduction: Thus, the IR absorption ... are not affected by plasmonic effects. I guess plasmonic resonances are meant here? Please clarify.

Answer: We updated the text, we meant indeed plasmonic resonances.

Reviewer: Add the technical details of the used mesh filter.

Answer: We added the details in the Materials and Methods.

Reviewer: Caption Figure 1: high values can cause instabilities. Comment: Maybe the signal is saturated?

Answer: As shown in the **SI Fig. 3** in combination with the **SI video 1-2**, the signal is unstable also if not saturated.

Reviewer: Figure 2 b: as it can be barely identified or mistaken please mark the measurement line by another colour, e.g. green.

Answer: We have update **Fig. 2**.

Reviewer: What means a 20 μ M of apoferritin; solution?

Answer: Yes, this is the concentration of protein in solution.

Reviewer: Please control referencing of Ref. [32] after (Figure 4b) in the Results section.

Answer: We updated the references.

Reviewer: A reference for the assignments used in the deconvolution of the amide I band (supplementary part) is missing.

A reference for the assignments used in the deconvolution of the amide I band (supplementary part) is missing.

Answer: We added the references to the assignments (*Ruggeri, Nature Comm., 2015; Shimanovich, Nature Comm., 2017*).

Reviewer: Supplementary: - 1st sentence: typo: ... we aimed at we aimed

Answer: Corrected.

Reviewer: - Line 33: 10 mW is given as the lower limit for typical power of the laser. This sounds quite high for the study of an organic samples.

Answer: We refer to the typical powers of QCLs lasers installed in commercial nanoIR and AFM-IR instruments.

Reviewer: Line 56: contact resonance at higher angle?

Answer: Corrected, "higher stiffness".

Reviewer 3

Reviewer: I don't recommend its publication in nature communication: it's not innovative enough and the format is not appropriated to this technical article. Here are some reasons and the list is not comprehensive. The authors introduce off-resonance, low power and short pulse AFM-IR to acquire infrared absorption spectra of thyroglobulin at the single molecule level, with a high signal-to-noise ratio. Without any doubt, it's essential to the community to have access to new protocols to improve the sensitivity and reliability of this technique but it's not innovative enough to be published in a journal like nature communication. I don't want to minimize the interest of this publication. Since the analysis of single protein using AFM-IR is a real challenge, I suggest to the authors to publish it in a more specialized journal where they can bring a real discussion, develop the different ambiguous points and illustrate it with more figures. This way, I think it should be appreciated to its full value by giving more details (by merging the main text and the supplementary documents) and complementary information and by highlighting the results with more proteins.

Answer: We would like to thank the reviewer for highlighting the challenge in reaching single protein absorption spectroscopy. We would like to highlight that the major objective and achievement of our study is the determination of protein structure at the single molecule scale.

Our study demonstrate that we can and bring together single protein molecule sensitivity with quantitative structural characterization from absorption spectroscopy. This achievement is not purely technical and it is not limited to the AFM-IR field. The characterisation of the chemical and structural properties of biomolecules at the single molecule scale opens a new fundamental window of observation of biomolecular process with a wide range of application in biochemistry, biophysics and biomedical research.

Following the comments of the Reviewer 3, as well as the other reviewers, we have updated the main text, Materials and Methods and SI to include a complete description of the specific technical details of our study.

Reviewer: You said "To date, however, the sensitivity of AFM-IR has been limited to the measurements of large (>0.3 μm) and flat (~2-10 nm) self-assembled monolayers or biomolecular aggregates composed of hundreds of molecules and has not demonstrated the ability to characterise single molecules": check other publications and update your bibliography.

Answer: As suggested also by **Reviewer 2**, there is not available in literature any study addressing the challenge to measure infrared absorption and quantitatively determine the secondary structure of a single protein by photothermal induced absorption spectroscopy.

Reviewer: -Figure 1 d) is really difficult to understand. Are you sure of your legend (blue dots 100ns? red dots 7%) I don't understand the link.

Answer: We updated and clarified the caption and the figure by using two different scales, we have also colour-coded the axes.

Reviewer: -Figure e): if your system is well optimized (in term of power and pulse width) the amplitude of the resonance of the substrate should be lower than the one of the protein. It's not the case. Why?

Answer: We observed a reproducible decrease of the amplitude of the resonance of the protein compared to the substrate, not only at the second resonance of oscillation of the cantilever, but also at the third resonance (**new SI Fig. 2**). This phenomenon likely occurs because it is easier to create a stable nanomechanical contact and resonance between the cantilever with the flat and stiff substrate than with a protein with smaller radius than the tip, soft and with spherical geometry.

Reviewer: In the caption of the figure 1 "(e) IR amplitude as a function of frequency pulse for signal on gold and on a thyroglobulin molecule at a laser power of 7% and at 1655 cm^{-1} " the power used for the experiments is 7 % but in the text it's written p5 that "already at the low laser power of 0.5 mW (1-5% of commercial IR sources), the enhancement of the field causes non-linearity and instabilities" maybe I have missed something but the power used for the experiments should be lower than 7% ? I suppose it's 7% of the power already reduce after the addition of the filter. I think it's better to use only mW and not %.

Answer: The reviewer has correctly interpreted the text, we used a laser power lower than 7% after introducing the mesh filter. Following the comment of the reviewer, we have clarified the text and express all the power measurements in mW.

Reviewer: Then figure 2 and 3 no power nor pulse width is indicated.

Answer: Spectra and maps were acquired with power between 0.13-0.35 mW and with a pulse width between 40-100 ns. It is now clarified in the caption of the figures, as well as in the main text and SI.

Reviewer: What about the discrepancies observed between FTIR results and AFM-IR? For example, why is your amide II lower? May be there is a physical reason behind it: discuss it.

Answer: The differences (shift~2-3 cm^{-1} shifts) observed between FTIR and AFM-IR spectra are smaller than our spectral resolution after smoothing (~4-6 cm^{-1}) and thus within experimental error. Furthermore, they can be related to the different instrumentation used, substrates and differences related to the comparison of a single protein (AFM-IR) and a population of proteins (FTIR).

Even if we normalised the spectra by the QCL emission profile, the smaller value of the Amide II mode is related to the low laser power in this region (**Supplementary Fig. S1**). The laser power in the Amide II is three times lower than in the Amide I, as well as the Amide II is absorbing IR light approximately twice less than the Amide I, thus the detection of Amide II is at the limit of instrumental sensitivity. While, it has been recently reported that for ordered biomaterials the lower absolute value of this band could be attributed to polarisation effects (*Giliberti, Nano Letters, 2019*), we believe polarisation effects are not likely for isotropic proteins.

Reviewer: But my main concern is about the interpretation of the local IR spectra: “Apoferitin has an α -helical content of 75% (Figure 4a),³⁴ while thyroglobulin has major contributions from α -helix and coils (55%), β -turn (17%) and intramolecular β -sheet structures (28%) (Figure 4b).” How you can propose such repartition without showing a curve fitting? the reference extract from FTIR curve fitting (reference) should be in the main document.

Answer: The curve fitting of the AFM-IR data was already explained and illustrated in the **old SI Fig. 6** and in the Materials and Methods. The procedure of calculation of the secondary structure has been identically undertaken for both proteins and for the spectra that we obtained by AFM-IR and FTIR, and based on protocols widely accepted in literature (*Ruggeri, Nature Communications, 2015*; *Shimanovich, Nature Communications, 2017*). Our determination of secondary structure is in excellent agreement with the structures of Apoferitin and Thyroglobulin determined previously in literature by FTIR and X-ray crystallography.

REVIEWERS' COMMENTS:

Reviewer #2 (Remarks to the Author):

The manuscript was revised very well.
I recommend to accept it for publication.

technical comments:

- The authors should consider to check if the maximum of signal as given in the text in connection with Figure 1 f) is at \sim -2kHz off-resonance.
- Fig S4; "FTIR" is found in the inset and caption, but the axis label is "IR amplitude"?

Reviewer: 2

Reviewer: The manuscript was revised very well. I recommend to accept it for publication.

Answer: We would like to thank the reviewer for these very positive comments on the quality and significance of our study.

Reviewer: The authors should consider to check if the maximum of signal as given in the text in connection with Figure 1 f) is at \sim -2kHz off-resonance.

Answer: We thank the reviewer for the comment. Indeed, the maximum is at -1.6 kHz, even if the response is flat within the error in the range from -1.4 to -1.8 kHz. We modified the text accordingly

Reviewer: Fig S4; "FTIR" is found in the inset and caption, but the axis label is "IR amplitude"?

Answer: Corrected to "IR Absorption".